# A Phenylacetamide Resveratrol Derivative Exerts Inhibitory Effects on Breast Cancer Cell Growth

**DOI:** 10.3390/ijms22105255

**Published:** 2021-05-17

**Authors:** Adele Chimento, Anna Santarsiero, Domenico Iacopetta, Jessica Ceramella, Arianna De Luca, Vittoria Infantino, Ortensia Ilaria Parisi, Paola Avena, Maria Grazia Bonomo, Carmela Saturnino, Maria Stefania Sinicropi, Vincenzo Pezzi

**Affiliations:** 1Department of Pharmacy and Health and Nutritional Sciences, University of Calabria, Via Pietro Bucci, Arcavacata di Rende, 87036 Cosenza, Italy; domenico.iacopetta@unical.it (D.I.); jessicaceramella@gmail.com (J.C.); ariannadl@hotmail.it (A.D.L.); ortensiailaria.parisi@unical.it (O.I.P.); paox1982@hotmail.it (P.A.); s.sinicropi@unical.it (M.S.S.); 2Department of Science, University of Basilicata, Viale dell’Ateneo Lucano 10, 85100 Potenza, Italy; anna.santarsiero@unibas.it (A.S.); vittoria.infantino@unibas.it (V.I.); mariagrazia.bonomo@unibas.it (M.G.B.); 3Spinoff TNcKILLERS, Viale dell’Ateneo Lucano 10, 85100 Potenza, Italy

**Keywords:** resveratrol, phenylacetamide RSV derivatives, breast cancer cell lines, antiproliferative activity, anti-inflammatory activity, cell cycle arrest, cell death

## Abstract

Resveratrol (RSV) is a natural compound that displays several pharmacological properties, including anti-cancer actions. However, its clinical application is limited because of its low solubility and bioavailability. Here, the antiproliferative and anti-inflammatory activity of a series of phenylacetamide RSV derivatives has been evaluated in several cancer cell lines. These derivatives contain a monosubstituted aromatic ring that could mimic the RSV phenolic nucleus and a longer flexible chain that could confer a better stability and bioavailability than RSV. Using MTT assay, we demonstrated that most derivatives exerted antiproliferative effects in almost all of the cancer cell lines tested. Among them, derivative **2**, that showed greater bioavailability than RSV, was the most active, particularly against estrogen receptor positive (ER+) MCF7 and estrogen receptor negative (ER-) MDA-MB231 breast cancer cell lines. Moreover, we demonstrated that these derivatives, particularly derivative **2**, were able to inhibit NO and ROS synthesis and PGE2 secretion in lipopolysaccharide (LPS)-activated U937 human monocytic cells (derived from a histiocytoma). In order to define the molecular mechanisms underlying the antiproliferative effects of derivative **2**, we found that it determined cell cycle arrest at the G1 phase, modified the expression of cell cycle regulatory proteins, and ultimately triggered apoptotic cell death in both breast cancer cell lines. Taken together, these results highlight the studied RSV derivatives, particularly derivative **2**, as promising tools for the development of new and more bioavailable derivatives useful in the treatment of breast cancer.

## 1. Introduction

Many foods and their bioactive components, including polyphenols, are beneficial for various diseases. Several in vitro and in vivo studies have indicated that a diet based on the consumption of cereals, legumes, vegetables, and fruits containing high levels of polyphenols prevents many diseases, including cancer [1,2]. Among them, a promising candidate is resveratrol (3,5,4′-trihydroxystilbene) (RSV), a naturally occurring non-flavonoid polyphenol [3]. RSV is the major phytoalexin produced by plants in response to stress, injury, infection, or UV radiation and is mainly present in peanuts, grapes, red wine, and some berries [4]. Although some of these plants and their extracts have been used for therapeutic purposes by ancient cultures, RSV itself was first described in 1939, when it was isolated from *Veratrum grandiflorum* [5]. Chemically, RSV is a stilbene compound and consists of two aromatic rings hydroxyl substituted with a C=C bond between them. It exists as *cis* and *trans* isomeric forms, with *trans* to *cis* isomerization facilitated by UV exposure, during the fermentation of skins, or under high pH conditions [6]. The *trans* form has greater stability and biological activity than the *cis* form, and is responsible for the induction of cellular responses. The increased interest in the *trans* isoform is also closely related to epidemiological studies showing an inverse relationship between moderate wine consumption and cardiovascular disease (the so-called “French paradox”). Moreover in vitro and in vivo studies have also demonstrated beneficial effects of RSV on human health [7,8], even though the mechanisms have not yet been fully elucidated. RSV has been reported to exhibit numerous activities, including antioxidant [9], anti-inflammatory [10], cardioprotective [11], neuroprotective [12] and anticancer properties [13,14] and so on, by acting through several molecular mechanisms [7]. RSV is able to protect against oxidative stress mainly by: (i) reducing reactive oxygen species (ROS) generation; (ii) directly scavenging free radicals; (iii) improving endogenous antioxidant enzymes (e.g., superoxide dismutase (SOD), catalase (CAT), and glutathione reductase (GSH)); (iv) promoting antioxidant molecules and the expression of related genes involved in mitochondrial energy biogenesis, mainly through AMPK/SIRT1/Nrf2, ERK/p38 MAPK, and PTEN/Akt signaling pathways; and (v) inducing autophagy via the mTOR-dependent or TFEB-dependent pathway [7]. Furthermore, RSV exerts anti-inflammatory effects through the inhibition of cyclooxygenase-1 (COX-1), cyclooxygenase-2 (COX-2), and 5-lipoxygenase catalytic activity, and consequent suppression of prostaglandins, thromboxanes, and leukotriene formation [15]. RSV was found to reduce LPS-induced nitric oxide (NO) and tumor necrosis factor alpha (TNF-α) production in primary microglia [16], prevent LPS-induced microglial BV-2 cell activation [17], and inhibit prostaglandin E2 (PGE2) and free radical production by rat primary microglia [18], modulating inflammatory responses. Since inflammation is a critical component of tumor progression and plays a key role in the tumor microenvironment [19], RSV represents a promising candidate for cancer prevention and/or treatment. There is a large body of in vitro studies that have demonstrated the proapoptotic and antiproliferative actions of RSV in several tumors [13], including lymphoblastic leukemia [20], colon [21], pancreatic [22], melanoma [23], gastric [24], cervical [25], ovarian [26], endometrial [27], liver [28], prostate [29] and breast [30] cancers, which confirmed its antitumor properties. These cancer chemopreventive effects are also corroborated by preclinical in vivo studies and clinical trials [31,32,33].

Although several studies have confirmed the beneficial effects of RSV on health, its therapeutic application is limited due to its short biological half-life and rapid metabolism and elimination, which limit its systemic bioavailability [34,35]. Indeed, in enterocytes, RSV undergoes phase II of drug metabolism, producing polar metabolites, which are easily excreted from the body. Specifically, it is conjugated with sulfate and glucoronate, and after these reactions, RSV metabolites can: (a) be transported across the apical membrane and reach the intestinal lumen, where they can be processed by the intestinal microbiota generating dihydroresveratrol (DHR), lunularin (L), and 3,4′-dihydroxy-trans-stilbene; (b) pass through the basolateral membrane and enter the bloodstream through which, by binding to blood proteins, they reach other tissues, such as the liver, kidneys and other peripheral tissues [36].

In the last decade, in order to overcome the limitations of low water solubility, absorption, transport across the membranes, and poor bioavailability of RSV, new synthetic derivatives and bio-isosteric analogues have been prepared [14,37,38,39,40,41]. Some of these possess a stronger pharmacological potency and a better pharmacokinetic profile than RSV itself [38,39]. It has been reported that methoxylation increases the molecule’s lipophilicity, thus promoting cell permeability with a greater metabolic stability and bioavailability [37]. The beneficial properties of other RSV derivatives have also attracted increased interest in recent years [37]. RSV glycosylation, alkylation, halogenation, hydroxylation, methylation, and prenylation could lead to the development of new derivatives with enhanced bioavailability and pharmacological activity [37].

In this work, the inhibitory effects on cancer cell growth of a series of synthetic RSV phenylacetamide analogues, synthetized by some of us [42], have been evaluated. These derivatives possess a monosubstituted aromatic ring that can mimic the RSV phenolic nucleus and a longer flexible chain that can confer better stability and bioavailability than RSV (Figure 1). We demonstrated that most derivatives exerted antiproliferative effects in almost all of the cancer cell lines used, and inhibited LPS-induced ROS and NO production and PGE2 secretion. We found that derivative **2** displayed a higher bioavailability compared to RSV and was the most active in exerting anti-inflammatory and anti-proliferative effects; this derivative was able to induce cell cycle arrest and apoptosis of estrogen receptor positive (ER+) MCF7 and estrogen receptor negative (ER-) MDA-MB231 breast cancer cells.

## 2. Results

### 2.1. Phenylacetamide RSV Derivatives Display Antiproliferative Effects on Several Cancer Cell Lines

The antiproliferative effects of phenylacetamide RSV derivatives were evaluated in several cancer cell lines, testing a wide range of concentrations from 2.5 to 40 µM (Figure 2). We observed that derivative **1** causes a significant reduction only in MDA-MB231 and U937 cell viability, starting from the lowest dose of 5 µM (Figure 2A); starting from the same dose, derivatives **2** (Figure 2B) and **6** (Figure 2F) exhibited inhibitory effects in the MCF7 cell line. Moreover, derivative **2** displayed significant cytotoxic action in MDA-MB231 (starting from the 10 µM dose), in U937 (starting from the 2.5 µM dose), and in H295R cells (only at the highest dose of 40 µM) (Figure 2B). The same higher dose (40 µM) of derivative **3** reduced MDA-MB231, H295R, and U937 cell viability (Figure 2C). The 10 µM dose of derivatives **3** (Figure 2C), **4** (Figure 2D), and **5** (Figure 2E)**,** and the 20 and 40 µM doses of derivative **4**, inhibited the MCF7 and MDA-MB231 and U937 cell viability, respectively. At the latter higher concentrations, derivatives **5** (Figure 2E) and **6** (Figure 2F) exerted inhibitory effects on MDA-MB231 (derivative **5**) and R2C and U937 (derivative **6**) cells. Interestingly, only derivatives **2** (Figure 2B) and **1** (Figure 2A) did not exhibit significant cytotoxic effects in the normal mouse fibroblast immortalized 3T3L1 cell line, whereas all other derivatives tested, even at different percentages, showed inhibitory effects. Moreover, we wanted to compare the effects of the same concentration ranges of derivative **2** with those of RSV on the proliferative behavior of tumor cells. The results reported in Appendix A show that RSV exhibited less inhibitory activity than derivative **2**, particularly in breast cancer cells. Moreover, derivative **2** did not show an antiproliferative effect on 3T3L1 cells, as RSV does (Appendix A).

### 2.2. Phenylacetamide RSV Derivatives Decrease LPS-Induced NO and ROS Production

It has been reported that inflammation is associated with tumorigenesis. Immune cells and inflammatory cells infiltrate solid tumors, influencing most tumorigenesis stages [19,43]. It is known that LPS induces an inflammatory state characterized by the release of pro-inflammatory mediators such as ROS and NO, two well-characterized chemokines that play a key role in inflammation [44,45]. RSV and its phenylacetamide derivatives were evaluated for their ability to affect NO and ROS levels in LPS-activated U937 cells (Figure 3B,C and Appendix A). Since no effects on cell viability were shown for RSV and all derivatives at doses lower than 0.1 µM in U937 cells (Figure 3A and Appendix A), we chose this dose as the optimal concentration to evaluate the anti-inflammatory activity of the phenylacetamide RSV derivatives, and we used RSV as a reference compound. U937 cells were differentiated into macrophages with PMA and pre-incubated for 1 h with DMSO or phenylacetamide RSV derivatives at 0.1 µM. Then, macrophages were induced with 400 ng/mL LPS. Following 24 h incubation at 37 °C, NO and ROS levels were detected. LPS induced a marked increase in NO (Figure 3B and Appendix A) and ROS production (Figure 3C and Appendix A) in PMA-treated U937 cells. The majority of the tested derivatives inhibited LPS-induced NO and ROS synthesis to various degrees. Derivative **6** was the most active in reducing ROS levels (by about 40%) (Figure 3C). Derivatives **1** (Figure 3C) and **2** (Figure 3C and Appendix A) were also able to lower ROS production in LPS-activated macrophages. All tested phenylacetamide RSV derivatives significantly decreased NO levels. However, the best results were obtained with derivatives **2**, **4**, and **5,** which abolished the LPS-induced increase (Figure 3B). In accordance with previous findings [46], RSV was not able to significantly reduce NO at the tested concentration (Appendix A). Moreover, derivative **2** was more effective than RSV in lowering ROS in this LPS-activated macrophage cell line (Appendix A).

### 2.3. Phenylacetamide RSV Derivatives Reduce LPS-Triggered PGE2 Secretion

Taking into consideration the putative anti-inflammatory proprieties of the phenylacetamide RSV derivatives, we wondered whether they were able to reduce the secretion of PGE2, a potent inflammatory mediator derived from arachidonic acid by the activity of cyclooxygenase-2. To this end, U937/PMA cells treated with 0.1 µM of RSV or derivatives **1**–**6** were activated with LPS for 48 h and PGE2 levels were measured in cell culture media. All phenylacetamide RSV derivatives, except derivative **1**, lowered PGE2 secretion (Figure 3D). The most active derivative was **2**, which reduced PGE2 levels by about 30% when compared to the LPS-treated cells (Figure 3D). Notably, derivative **2** was more active than RSV in diminishing PGE2 secretion following the LPS activation of macrophages (Appendix A).

### 2.4. Derivative **2** Determines Cell Cycle Arrest at G1 Phase in Both Human MCF7 and MDA-MB231 Breast Cancer Cell Lines

In order to better investigate the molecular mechanism underlying the antiproliferative effects of derivative **2** in ER+ MCF7 and ER- MDA-MB231 breast cancer cells, we performed cell cycle analysis using flow cytometry (Figure 4). Cell cycle changes produced by this RSV derivative were measured and the DNA histograms are reported in Figure 4. We observed a significant increase in the G1 cell population after 24 h for the 20 and 40 µM doses, accompanied by a variable decrease in the S and G2 phase cells, compared to the control, in derivative **2**-treated cells. These results confirmed a cell cycle arrest in the G1 phase in both MCF7 (Figure 4A) and MDA-MB231 (Figure 4C) cells. To explore the molecular alterations that might underpin cell cycle arrest, changes in the expression of specific cell cycle regulatory proteins were determined by Western blot analysis. Our results showed that treatment with derivative **2** caused a dose-dependent decrease in cyclin D1 (CCND1) and cyclin-dependent kinase 4 (CDK4) protein expression in MCF7 (Figure 4B) and MDA-MB231 (Figure 4D) cells. These results confirmed that exposure to this derivative invokes a signaling cascade that culminates in cell cycle arrest.

### 2.5. Derivative **2** Induces Gross Morphological Changes and Apoptosis in Both Human MCF7 and MDA-MB231 Breast Cancer Cell Lines

To further characterize whether the growth inhibitory activity of derivative **2** on MCF7 and MDA-MB231 cells was related to the induction of cell death, we examined its effects on cell morphology and apoptosis. We found that derivative **2** determines a marked decrease in cell number and gross morphological changes in MCF7 (Figure 5A) and MDA-MB231 cells (Figure 5D); these are dying cells, in which the 20 and 40 µM doses clearly determine an aberrant morphology as compared with the untreated cells (vehicle only) and the presence of many round floating cells.

The ability of derivative **2** to trigger apoptosis in human MCF7 and MDA-MB231 breast cancer cell lines was confirmed by the evaluation of DNA fragmentation. TUNEL staining demonstrated the presence of increased positive cells following derivative **2** treatment (Figure 5B,E). In addition, DAPI staining evidenced that the untreated breast cancer cells had round nuclei with regular contours, while nuclei from cells treated with derivative **2** appeared shrunken and irregularly shaped or degraded with condensed DNA. Apoptosis regulation requires the involvement of proteins that belong to the bcl-2 family; this proteins family includes both proapoptotic and antiapoptotic members, such as bax and bcl2, respectively [47]. Using Western blot analysis, we found that the presence of derivative **2** increased bax expression and decreased bcl-2 in both MCF7 (Figure 5C) and MDA-MB231 (Figure 5F) cells. Particularly, in MDA-MB231 cells, this derivative determined p21 bax cleavage to the p18 form, which may serve to increase the intrinsic cytotoxic properties of bax and enhance its cell death function in the mitochondria [48]. We also detected the cleavage of parp1, which is considered one of the most important biochemical features of cells undergoing apoptosis [49]. Treatment with derivative **2** (20–40 μM) induced parp1 cleavage in both cell lines (Figure 5C,F). These results confirmed that in the studied human breast cancer cell lines, derivative **2** determines the induction of the apoptotic mechanism.

### 2.6. In Vitro Bioavailability of Derivative **2**

The in vitro bioavailability of derivative **2** was investigated using simulated gastrointestinal digestion according to the dialysis tubing procedure. The obtained results were expressed as bioavailability percentage, calculated as the percentage of the tested compound recovered in the bioaccessible fraction after in vitro gastric and intestinal digestions compared to the initial undigested sample (Equation (1)):Bioavailability (%) = (bioaccessible content)/(total undigested content) × 100(1)

After the first 2 h of simulated gastric digestion, derivative **2** bioavailability was equal to 23.1 ± 1.1%, while 36.3 ± 0.6% was recovered from the bioaccessible fraction after a further 4 h of simulated intestinal digestion (Table 1). Therefore, the cumulative bioavailability of the tested derivative after the two sequential digestion steps was around 59%. The obtained results showed an increased in vitro bioavailability of derivative **2** compared to RSV, characterized by a bioavailability of 13.4 ± 0.7% and 23.3 ± 0.6% after gastric and intestinal digestions, respectively, reaching a cumulative bioavailability of around 36% (Table 1).

## 3. Discussion

Amongst the phytochemicals present in food that exhibit considerable prospects in the treatment and management of human diseases, RSV, a stilbene-type aromatic phytoalexin, is one of the most widely studied [7]. In the last two decades, further studies have aimed to elucidate the mechanisms underlying RSV activity in reducing cancer initiation, promotion, and progression, as well as the occurrence of inflammation in various cancer models [7,13,14]. By interfering with diverse signal transduction pathways, RSV has been shown to control cell growth and division, apoptosis, metastasis, and angiogenesis [13]. To date, experimental in vitro and in vivo data and several clinical trials have demonstrated the potential role of RSV as a preventive or therapeutic agent for a wide range of cancers [13,20,21,22,23,24,25,26,27,28,29,30,31,32,33]. This is due to its ability to modulate different targets and act on different pathways that are usually altered in cancer, together with a low toxicity [50]. Moreover, the combined treatment of RSV with diverse chemotherapeutics revealed enhanced antineoplastic effects [51]. Even though human clinical trials have produced positive results, some critical concerns, partly due to the dosing protocols and low bioavailability, have arisen [8,34,35]. To overcome these limitations, numerous efforts focused on the formulation of delivery systems and the development of more bioavailable derivatives have been made [37]. From this point of view, we designed and synthesized a series of phenylacetamide RSV analogues that are characterized by a monosubstituted aromatic ring, mimicking the RSV phenolic nucleus, and a longer flexible chain that could confer higher stability and bioavailability. Using a panel of cancer cell lines, we evaluated the anticancer activity of these RSV analogues and found that almost all possessed anticancer activity. Particularly, derivative **2** was the most effective in reducing cancer cell viability, together with a lack of cytotoxicity against the murine fibroblast 3T3L1 cell line, which was used as a normal cell model. Moreover, it displayed enhanced inhibitory activity against estrogen receptor positive (ER+) MCF7 and estrogen receptor negative (ER-) MDA-MB231 breast cancer cell growth and had a higher bioavailability of 23% compared to RSV.

Next, we wondered whether derivative **2** could also act as modulator of inflammatory responses in U937 cells differentiated into macrophages. Indeed, it is noteworthy that inflammation is usually associated with the induction or the exacerbation of about a quarter of cancer cases, very quickly resulting in a chronic situation that leads to higher destruction and lower healing of the affected tissue(s) [43]. Once again, RSV has been shown to modulate pro-inflammatory proteins or their signal transduction pathways, as well as other pathways producing precursors of inflammation [7,10,11]. Our outcomes clearly indicate that derivative **2**, when used at a concentration that is not toxic for the adopted cell model (viz. 0.1 µM), is able to reduce NO synthesis and PGE2 levels (released in the culture medium) in U937/PMA cells exposed to LPS, by about 40% and 30%, respectively. The other analogues shared a similar anti-inflammatory activity, with a certain inter-series variability, but to a lesser extent. The anti-inflammatory activity synergizes with another property of derivative **2,** its ability to significantly decrease ROS levels in the same cellular context previously adopted, even though the best antioxidant performance was obtained with the use of derivative **6**. Taken together, these data proved that the lead derivative **2** possesses the ability to decrease cancer cell viability and diminish, contemporaneously, two important mediators of inflammation and ROS production. Next, we performed flow cytometry and Western blot analysis to further study the mechanisms involved in the observed anticancer activity of derivative **2**. Cell cycle analysis performed on the two breast cancer cell lines revealed a block at the G1 phase after exposure at the indicated doses of derivative **2**. These results have been also confirmed by Western blot analysis. A dose-dependent decrease in the CCND1 and CDK4 protein expression, in both MCF7 and MDA-MB231 cells, has been detected. DNA damage most likely occurs as a result of derivative **2** treatment; indeed, the role of the G1 checkpoint in limiting new cancer cell formation, with damaged DNA, is irrefutable nowadays, because alterations or mutations in several proteins involved in the next phase transition have been linked to specific tumors [52]. If the cells arrive in the G1 phase with damaged DNA, in one way or another, they sense this damage and send signals that cause alterations in the cyclins and CDKs that “put the brake on”, producing a block in G1 phase until the DNA damage is repaired. G1-phase arrest of cell cycle progression gives cells the ability to repair damage or die from apoptosis [52,53]. Therefore, we studied the mechanism by which breast cancer cells, blocked at the G1 phase, died following derivative **2** treatment. The morphological cell changes observed are, undoubtedly, a signal of cell death [54]; indeed, derivative **2** is able to induce apoptotic mechanisms, with DNA damage, as determined by the means of TUNEL and parp1 cleavage assays. Moreover, we observed the increased expression of the pro-apoptotic protein bax, accompanied by a decrease in bcl2 levels. These proteins are the major members of the bcl-2 family that play a key role in tumor progression and inhibition of the intrinsic apoptotic pathway [47]. Therefore, the balance between pro- and anti-apoptotic members of this family determines the cellular fate; in this case, the increased bax/bcl2 ratio confirmed the apoptosis of breast cancer cell lines that was previously observed. It should be highlighted that the downregulation and upregulation of bcl2 and bax expression, respectively, triggered by derivative **2** are important markers that have already been clinically associated with better survival in patients affected by some types of tumors [55,56,57]. Thus, closing the circle, the latter data strengthen our previous cell cycle results, indicating that, following derivative **2** exposure, breast cancer cells are blocked at the G1 phase of the cell cycle, due to their inability to repair DNA damage, and are directed toward cell death by the apoptotic mechanism. In conclusion, the revealed features of this lead derivative make it a valuable molecule that is able to modulate several biological aspects of cancer cell growth, shedding light on the research into new and more active RSV derivatives.

## 4. Materials and Methods

### 4.1. Cell Cultures and Treatments

MCF7 and MDA-MB231 breast cancer cells, H295R adrenocortical cancer cells, R2C rat Leydig tumor cells, and 3T3L1 mouse embryonic fibroblast cells were purchased from the American Type Culture Collection (ATCC) (Manassas, VA, USA). MCF7 cells (ER positive breast cancer cells) were maintained as previously described [58,59]. MDA-MB231 cells (ER, PR, and HER2 triple negative breast cancer cells) were maintained in Dulbecco’s Modified Eagle Medium/F12 (DMEM/F12) medium supplemented with 10% fetal bovine serum (FBS), 1% L-glutamine (L-Glu), and 1% penicillin/streptomycin (P/S) (Sigma-Aldrich, Milano, Italy) (complete medium). H295R adrenocortical cancer cells were cultured in DMEM/F12 medium supplemented with 1% Insulin-Transferrin-Selenium (ITS) Liquid Media Supplement, 10% FBS), 1% L-Glu, and 1% P/S (complete medium). R2C rat Leydig tumor cells were maintained as previously described [60]. The 3T3L1 mouse embryonic fibroblast cell line, obtained from ATCC, was cultured in DMEM with phenol red supplemented with 10% FBS, 1% L-Glu, and 1% P/S (Sigma-Aldrich) (complete medium). The human pro-monocytic cells derived from a histiocytoma (U937 cells) were purchased from the Interlab Cell Line Collection (ICLC) (IRCCS AOU San Martino IST) and grown in Roswell Park Memorial Institute (RPMI) 1640 medium supplemented with 10% fetal bovine serum, 2 mM L-Glu, 100 U/mL penicillin, and 100 μg/mL streptomycin. All cells were maintained at 37 °C in a humidified atmosphere of 95% air and 5% CO_2_ and were screened periodically for Mycoplasma contamination. U937 cells were differentiated to macrophages by adding 10 ng/mL phorbol 12-myristate 13-acetate (PMA, Sigma-Aldrich) for 16 h. Where indicated, U937/PMA cells were stimulated for 24 or 48 h with 400 ng/mL of lipopolysaccharides from *Salmonella enterica* serotype typhimurium (LPS, Sigma-Aldrich) after one hour of treatment with the phenylacetamide RSV derivatives **1**–**6**. All cell lines were treated in complete medium at the times and concentrations indicated with phenylacetamide RSV derivatives (**1**–**6**) or RSV.

### 4.2. Cell Viability Assay

Cell viability was determined as previously described [61] by colorimetric 3-(4,5-dimethylthiazol-2-yl)-2,5-diphenyltetrazolium bromide (MTT) assay, which measures mitochondrial activity in viable cells. Specifically, the cells (2.5 × 10^4^) were plated in a 48-well plate and, after 48 h, were treated with phenylacetamide RSV derivatives (**1**–**6**) at increasing concentrations (2.5, 5, 10, 20, and 40 µM) for 72 h. Control groups were treated with Dimethyl Sulfoxide (DMSO) (Sigma-Aldrich) equal to the highest percentage of (<0.1%) solvent used in the experimental conditions for the MTT assay. After treatment, fresh MTT, resuspended in phosphate buffered saline (PBS), was added to each well (final concentration 0.5 mg/ml) and the plate was incubated at 37 °C for 3 h in a humidified 5% CO_2_ incubator. After incubation, media were removed, and formazan crystals were dissolved in 200 µL of DMSO (Sigma-Aldrich) for 10 min with gentle agitation. Each experiment was performed with six replicates three times; the optical density was measured at 570 nm with a spectrophotometer (Synergy H1 plate reader, BioTek Instruments, Inc., Winooski, VT, USA).

### 4.3. Flow Cytometric Analysis of DNA Content

MCF7 and MDA-MB231 breast cancer cells (5 × 10^5^/well) were seeded in six multi-well plates for 24 h. Sub-confluent monolayers were depleted of serum for 12 h and treated for an additional 24 h with derivative **2** (20 and 40 µM). The cells were harvested by trypsinization and resuspended with 0.5 mL of DNA staining solution (0.1 mg/mL propidium iodide (PI), 0.1% sodium citrate, and 0.1% Triton X-100, 0.02 mg/mL RNase; Sigma). The DNA content was measured using a CytoFLEX flow cytometer (Beckman Coulter SRL, Milano, Italy). Nuclei (10,000 events) were analyzed from each sample. The percentage of cells in the G1, S and G2/M phases of the cell cycle were determined by analysis with CytExpert software (Beckman Coulter SRL, Milano, Italy).

### 4.4. Western Blot Analysis

Total proteins were subjected to Western blot analysis [60]. Blots were incubated overnight at 4 °C with antibodies against cyclin D1 (CCND1), cyclin-dependent kinase 4 (CDK4), bax, bcl2, and parp1 (all from Santa Cruz Biotechnology, Santa Cruz, CA, USA). Membranes were incubated with horseradish peroxidase (HRP)-conjugated secondary antibodies (Amersham Pharmacia Biotech, Piscataway, NJ, USA) and immunoreactive bands were visualized with the ECL Western blotting detection system (Amersham Pharmacia Biotech). To ensure equal loading of proteins, membranes were stripped and incubated overnight with glyceraldehyde 3-phosphate dehydrogenase (GAPDH) antibody (Santa Cruz Biotechnology).

### 4.5. Phase Contrast Microscopy for Morphological Evaluation

MCF7 and MDA-MB231 cells were seeded in six-well plates at a density of 5 × 10^5^ cells/well for 24 h and then left untreated (0) or treated with derivative **2** (20 and 40 μM) for 24 h. Subsequent to treatment, culture plates were observed using an inverted phase contrast microscope (Olympus CKX53 inverted microscope, Waltham, MA, United States) and images were captured (×10 objective).

### 4.6. Terminal Deoxynucleotidyl Transferase-Mediated dUTP Nick-End Labelling (TUNEL) Assay

Apoptosis was detected using the TUNEL assay, according to the guidelines of the manufacturer (CF^TM^488A TUNEL Assay Apoptosis Detection Kit, Biotium, Hayward, CA, USA). The cells were grown on glass coverslips and then treated with derivative **2**. After paraformaldehyde fixation, the TUNEL reaction mixture containing the terminal deoxynucleotidyl transferase (TdT) enzyme was added, as reported by Iacopetta et al. [62,63]. Samples were washed, incubated with DAPI (Sigma; 0.2 µg/mL) for nuclei counterstaining, and then observed and imaged under a fluorescence microscope (Leica DM6000, Leica Microsystems GmbH, Wetzlar, Germany; magnification ×20, λ_ex/em_ maxima of 490/515 nm for CF^TM^488A or 350/460 nm for DAPI). Images are representative of three independent experiments.

### 4.7. ROS and NO Detection

For reactive oxygen species and nitric oxide analysis, U937 cells were differentiated in macrophages with PMA and then activated by LPS in the presence (or absence) of 0.1 µM of the phenylacetamide RSV derivatives. After 24 h, ROS and NO levels were measured using 6-carboxy-2′,7′-dichlorodihydrofluorescein diacetate (DCF-DA, Thermo Fisher Scientific, San Jose, CA, USA) and 4-amino-5-methylamino-2′,7′-difluorofluorescein diacetate (DAF-FM Diacetate, Thermo Fisher Scientific), respectively [64]. The fluorescence was analyzed on a microplate reader (Glomax, Promega, Madison, WI, USA).

### 4.8. PGE2 Quantification

U937/PMA cells were treated with phenylacetamide RSV derivatives (0.1 µM) for one hour, and then stimulated with 400 ng/mL LPS. After 48 h, the cell culture media were collected to measure PGE2 levels using the DetectX^®^ Prostaglandin E2 High Sensitivity Immunoassay Kit (Arbor Assays, Ann Arbor, MI, USA) according to the manufacturer’s instructions.

### 4.9. In Vitro Bioavailability Studies

In vitro bioavailability studies were carried out following the dialysis tubing procedure, which consisted of two sequential digestion steps involving pepsin and pancreatin, respectively [65]. Simulated gastric digestion was performed by preparing a digestion mixture consisting of 100 µL of a derivative **2** solution (10 mM in DMSO), 1.0 mL of HCl (0.85 N), and 3 mL of a sodium cholate solution (2% *w*/*v* in distilled water). Then, the mixture was placed into a dialysis bag (Spectrum Laboratories Inc., MWCO: 12,000–14,000 Dalton, GA, USA) and dialyzed against 10 mL of a 0.85 N HCl solution (pH 1.0) at 37 °C. After 2 h, a 2 mL aliquot was withdrawn and analyzed to evaluate the amount of derivative **2** in the bioaccessible fraction. For sequential intestinal digestion, 11 mg of amylase, 11 mg of esterase, and 1.3 mL of a 0.8 M NaHCO_3_ solution containing 22.60 mg porcine pancreatin/mL were introduced into the dialysis bag, which was then dialyzed against 10 mL of buffer solution at pH 7.0 for a further 4 h at 37 °C. Then, 2 mL of the external solution was withdrawn and analyzed to assess derivative **2** concentration. UV/Vis spectroscopy (UV/Visible Spectrophotometer Evolution 201, Thermo Fisher Scientific) was used to determine the concentration of derivative **2** in the samples obtained after simulated gastric and intestinal digestions. The in vitro bioavailability studies were carried out using RSV as a reference derivative. Experiments were performed in triplicate.

### 4.10. Statistical Analysis

All experiments were performed at least three times. Data are expressed as mean values ± standard error (SE). The statistical significance between control (basal, 0) and treated samples was analyzed using GraphPad Prism 5.0 software (GraphPad Software, Inc., San Diego, CA, USA). Control and treated groups were compared using the analysis of variance (ANOVA) with Bonferroni, Dunn’s, or Tukey’s post hoc tests. Differences were considered as significant when *p* < 0.05.

## Figures and Tables

**Figure 1 ijms-22-05255-f001:**
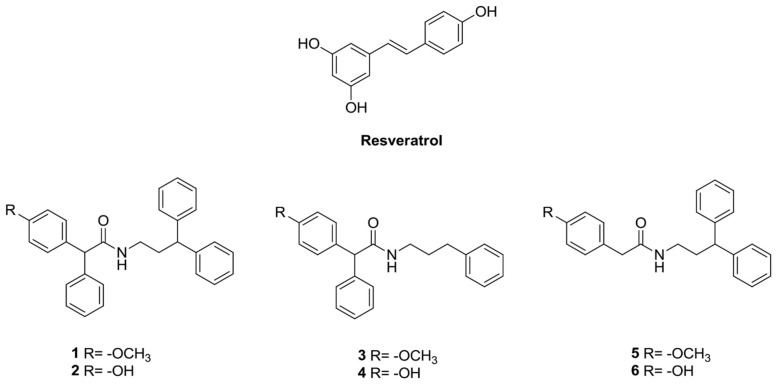
Molecular structure of RSV and the tested RSV derivatives (**1**–**6**).

**Figure 2 ijms-22-05255-f002:**
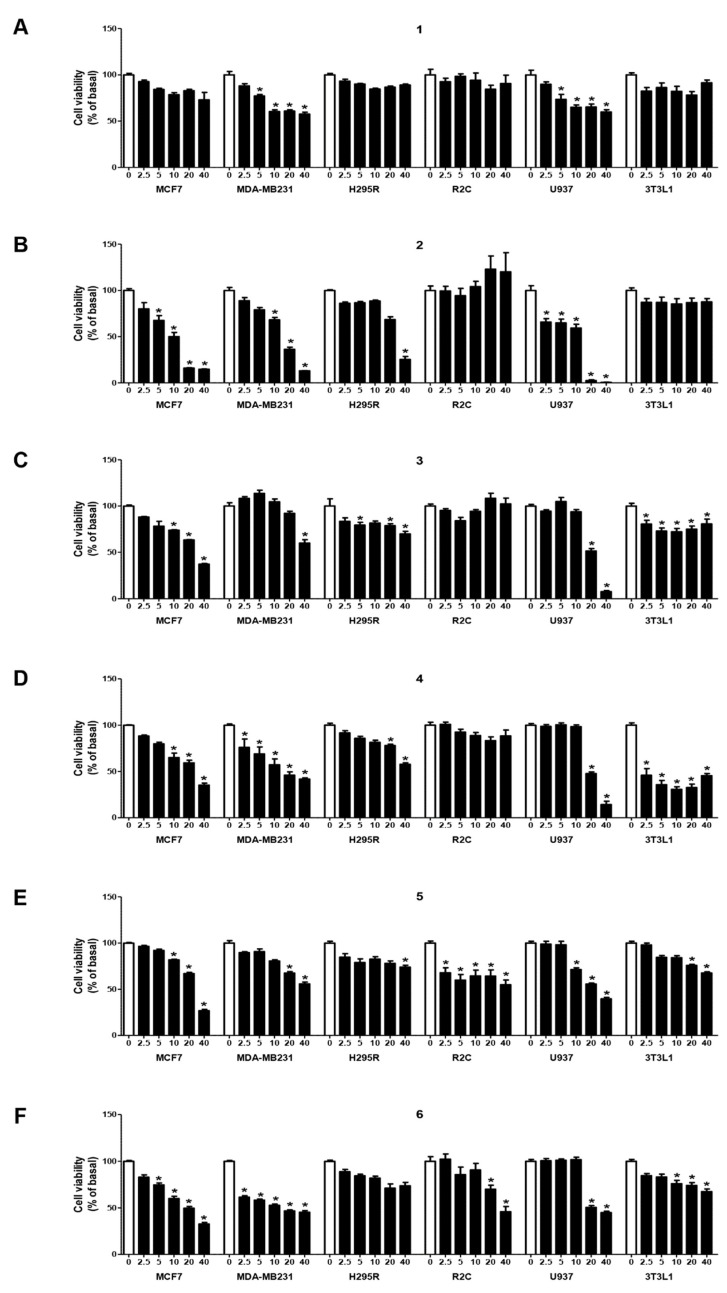
Effects of phenylacetamide RSV derivatives on cancer cell viability. MCF7, MDA-MB231, H295R, R2C, U937, and 3T3L1 cells were treated with vehicle DMSO (0) or the following phenylacetamide RSV derivatives: **1** (**A**), **2** (**B**), **3** (**C**), **4** (**D**), **5** (**E**), or **6** (**F**) at the indicated concentrations (2.5, 5, 10, 20, and 40 µM). Cell viability was assessed by MTT assay after 72 h exposure. Results are expressed as mean ± SE of three separate experiments (* *p* < 0.05 with respect to control (0)).

**Figure 3 ijms-22-05255-f003:**
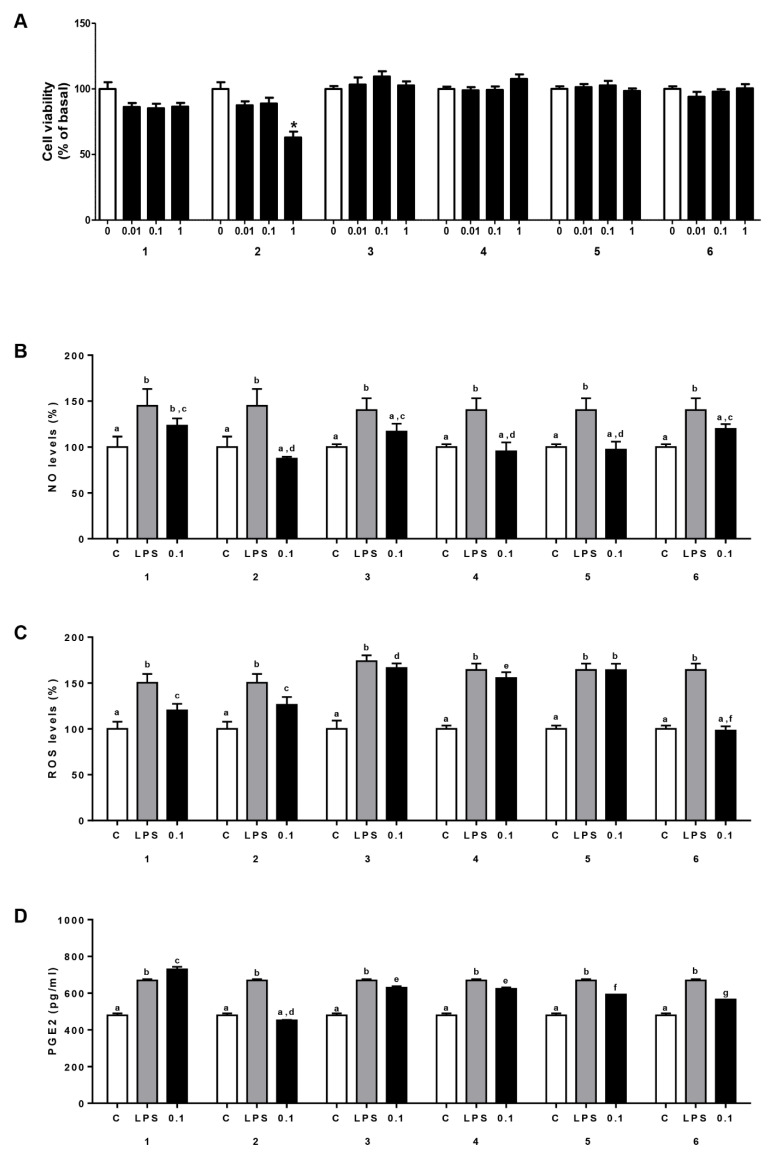
Effects of low doses of phenylacetamide RSV derivatives on U937 cell viability, NO and ROS levels, and PGE2 production. (**A**) U937 cells were treated with vehicle DMSO (0) or the phenylacetamide RSV derivatives (**1**–**6**) at the indicated concentrations (0.01, 0.1, or 1 µM). Cell viability was assessed by MTT assay after 72 h exposure. Results are expressed as mean ± SE of three separate experiments (* *p* < 0.05 with respect to control (0)). (**B**–**D**) In PMA-treated U937 cells, unstimulated (C) or activated with LPS alone (LPS), or in the presence of 0.1 µM of the phenylacetamide RSV derivatives (**1**–**6**), NO (**B**), ROS (**C**) and PGE2 (**D**) levels were quantified. Means ± SE of four independent experiments are shown. In (**B**–**D**), different letters indicate significant differences between treatments at *p* < 0.05 (Tukey’s post hoc test).

**Figure 4 ijms-22-05255-f004:**
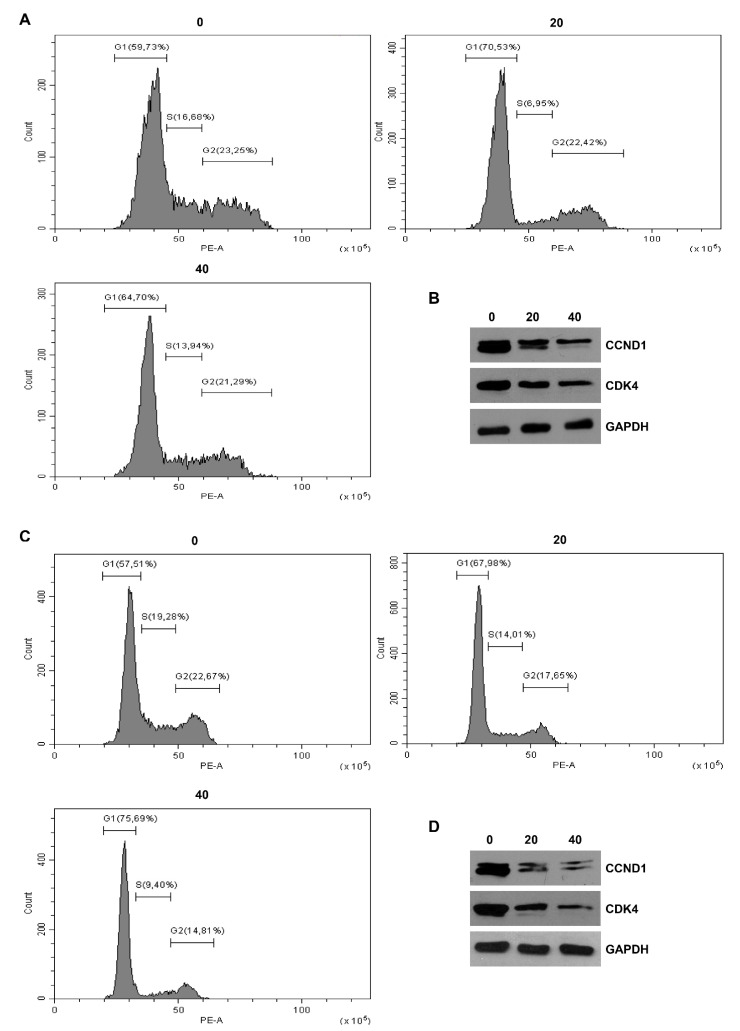
Effects of derivative **2** on the MCF7 and MDA-MB231 cell cycle distribution. (**A**,**C**) MCF7 (**A**) and MDA-MB231 (**C**) cells were synchronized in serum-free media for 12 h and then exposed to vehicle (0) or derivative **2** for 24 h at different concentrations (20 and 40 µM). The distribution of MCF7 and MDA-MB231 in the cell cycle was determined by flow cytometry using propidium iodide stained nuclei. (**B**,**D**) Western blot analysis of CCND1 and CDK4 was performed on equal amounts of total proteins extracted from MCF7 (**B**) and MDA-MB231 (**D**) cells treated with derivative **2** (20 and 40 µM) for 24 h. Blots are representative of three independent experiments with similar results. GAPDH was used as a loading control.

**Figure 5 ijms-22-05255-f005:**
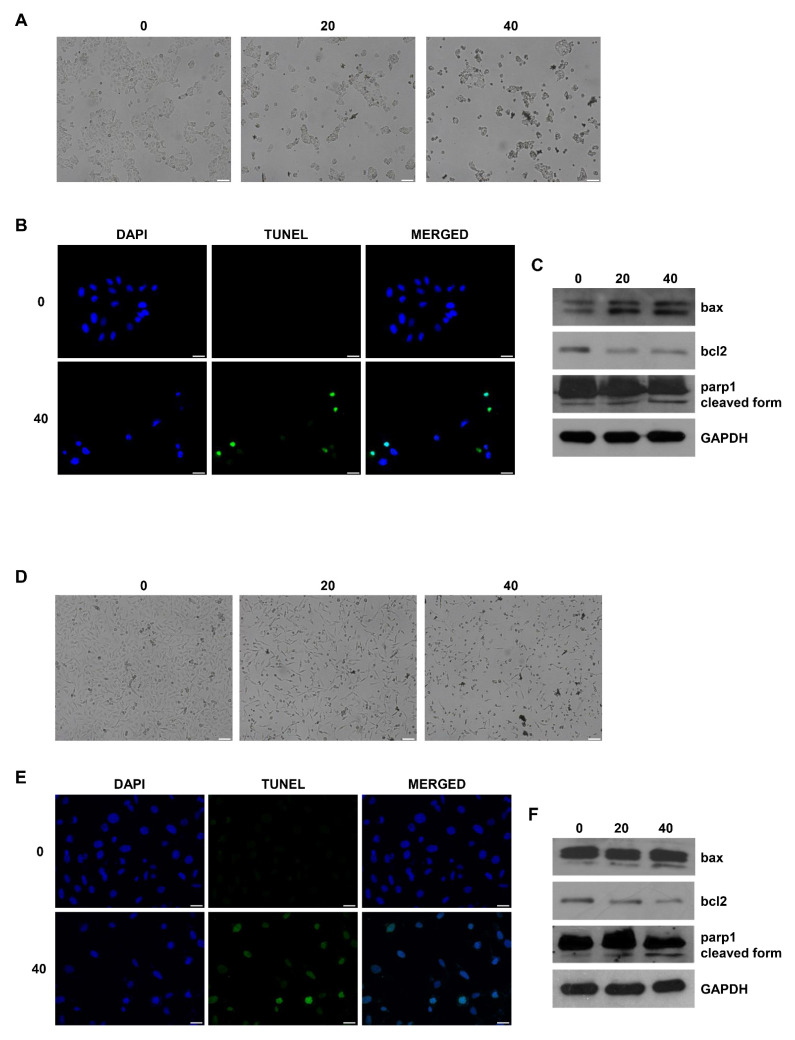
Effects of derivative **2** on MCF7 and MDA-MB231 cell morphology and apoptosis. MCF7 (**A**) and MDA-MB231 (**D**) cells were untreated (0) or treated with derivative **2** (20 and 40 μM) for 24 h; after treatment, cells were examined with a phase-contrast microscope (×10 objective). Images are from a representative experiment. Scale bar: 100 µm. (**B**,**E**) MCF7 (**B**) and MDA-MB231 (**E**) cells were untreated (0) or treated with derivative **2** (10 μM) for 72 h; after treatment, cells were fixed with paraformaldehyde and processed for TUNEL staining. Nuclei counterstaining was performed using DAPI. Fluorescent signals were observed under a fluorescent microscope (×20 objective). Images are from a representative experiment. Scale bar: 50µm. (**C**,**F**) MCF7 (**C**) and MDA-MB231 (**F**) cells were untreated (0) or treated with derivative **2** (20 and 40 μM) for 24 h. Western blot analyses of bax, bcl-2, and parp1 were performed on equal amounts of total proteins. Blots are representative of three independent experiments with similar results. GAPDH was used as a loading control.

**Table 1 ijms-22-05255-t001:** Bioaccessibility (%) of derivative **2** and RSV.

Sample	Bioaccessibility (%) after Gastric Digestion	Bioaccessibility (%) after Intestinal Digestion	Cumulative Bioaccessibility (%)
Derivative **2**	23.1 ± 1.1	36.3 ± 0.6	~59
RSV	13.4 ± 0.7	23.3 ± 0.6	~36

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
