# Peer review of "A Phenylacetamide Resveratrol Derivative Exerts Inhibitory Effects on Breast Cancer Cell Growth"

_ijms, 2021, doi:10.3390/ijms22105255_

Round 1

Reviewer 1 Report

The reviewer does not support the publication of this manuscript. 1) The activities of these derivatives are very low. 2) Why there is no positive control? 3) Compared with resveratrol, what are the advantages of these derivatives?

Reviewer 2 Report

The subject of the paper “ Phenylacetamide resveratrol derivatives exert inhibitory effects on cancer cell growth”, is appropriate for the publication in Int. J. Mol. Sci.  

Manuscript provides a new set of experimental data regarding the effects of resveratrol (RSV) derivates in several cancer cell lines. Authors demonstrated that derivatives exerted antiproliferative effects in almost all the cancer cell lines tested. The derivative 2 displayed the greatest effects across cell lines. Authors demonstrated that derivative 2, were able to inhibit ROS and NO synthesis and PGE2 secretion in lipopolysaccharide (LPS)-activated U937 human monocytic cells.  In addition, authors explored a possible the molecular mechanisms underlying the antiproliferative effects of derivative 2. The subject of the present research has relevant information regarding the RSV derivates and their anti-cancer effects. The information and interpretation of the findings need minor improvement, see specific comments.

  1. Pg3 Ln126-127 Results – statement “cytotoxic effects in the normal human fibroblasts immortalized 3T3L-1 cell line” should be corrected as the 3T3L-1 cell line is derived from mouse embryo!
  2. Pg10 Fig. 5 – microscope images of MCF-7 (5A) and MDA-MB-231 (5B) should be presented at higher resolution. All microscope images are missing resolution bars (?? um).

Round 2

Reviewer 2 Report

The subject of the paper “ Phenylacetamide resveratrol derivatives exert inhibitory effects on cancer cell growth”, is appropriate for the publication in Int. J. Mol. Sci.  

Manuscript provides a new set of experimental data regarding the effects of resveratrol (RSV) derivates in several cancer cell lines. Authors demonstrated that derivatives exerted antiproliferative effects in almost all the cancer cell lines tested. The derivative 2 displayed the greatest effects across cell lines. Authors demonstrated that derivative 2, were able to inhibitROS and NO synthesis and PGE2 secretion in lipopolysaccharide (LPS)-activated U937 human monocytic cells from histiocytoma.  In addition, authors explored a possible the molecular mechanisms underlying the antiproliferative effects of derivative 2. Authors improved manuscript according reviewers’ comments. It is recommended to accept manuscript for publication.